# Economic Evaluation of Direct Oral Anticoagulants Compared to Warfarin for Venous Thromboembolism in Thailand: A Cost-Utility Analysis

**DOI:** 10.3390/ijerph20043176

**Published:** 2023-02-11

**Authors:** Siwaporn Niyomsri, Mantiwee Nimworapan, Wanwarang Wongcharoen, Piyameth Dilokthornsakul

**Affiliations:** 1Department of Medical Services, Ministry of Public Health, Nonthaburi 11000, Thailand; 2Population Health Sciences, Bristol Medical School, University of Bristol, Bristol BS8 2BN, UK; 3Pharmaceutical Care Training Center (PCTC), Department of Pharmaceutical Care, Faculty of Pharmacy, Chiang Mai University, Chiang Mai 50200, Thailand; 4Center for Medical and Health Technology Assessment (CM-HTA), Department of Pharmaceutical Care, Faculty of Pharmacy, Chiang Mai University, Chiang Mai 50200, Thailand; 5Division of Cardiology, Department of Internal Medicine, Faculty of Medicine, Chiang Mai University, Chiang Mai 50200, Thailand

**Keywords:** venous thromboembolism, warfarin, direct oral anticoagulants, economic evaluation

## Abstract

Background: Direct oral anticoagulants (DOACs) have been used for venous thromboembolism (VTE) in Thailand. However, they have not been listed in the National List of Essential Medicines (NLEM). A cost-effectiveness analysis is needed to aid policymakers in deciding whether DOACs should be listed in the NLEM. This study aimed to assess the cost-effectiveness of DOACs for patients with VTE in Thailand. Methods: A cohort-based state transition model was constructed from a societal perspective with a lifetime horizon. All available DOACs, including apixaban, rivaroxaban, edoxaban, and dabigatran, were compared with warfarin. A 6-month cycle length was used to capture all costs and health outcomes. The model consisted of nine health states, including VTE on treatment, VTE off treatment, recurrent VTE, clinically relevant non-major bleeding, gastrointestinal bleeding, intracranial bleeding, post-intracranial bleeding, chronic thromboembolic pulmonary hypertension, and death. All inputs were based on a comprehensive literature review. The model outcomes included total cost and quality-adjusted life-years (QALYs) with a 3% annual discount rate. A fully incremental cost-effectiveness analysis and the incremental cost-effectiveness ratio (ICER) per QALY gained were calculated at a willingness-to-pay (WTP) of THB 160,000/QALY ($5003). The robustness of the findings was assessed using deterministic and probabilistic sensitivity analyses. Results: All DOACs were associated with a decreased risk of VTE recurrence and intracranial hemorrhage. In the base-case analysis, apixaban could increase 0.16 QALYs compared with warfarin. An ICER for apixaban was 269,809 Thai baht (THB)/QALY ($8437/QALY). Rivaroxaban had a better QALY than warfarin at 0.09 QALYs with an ICER of 757,363 THB/QALY ($23,682/QALY). Edoxaban and dabigatran could also increase by 0.10 QALYs with an ICER of 709,945 THB ($22,200) and 707,145 THB ($22,122)/QALY, respectively. Our probabilistic sensitivity analyses indicated that warfarin had a 99.8% possibility of being cost-effective, while apixaban had a 0.2% possibility of being cost-effective at the current WTP. Other DOACs had no possibility of being cost-effective. Conclusions: All DOACs were not cost-effective for VTE treatment at the current WTP in Thailand. Apixaban is likely to be the best option among DOACs.

## 1. Introduction

Venous thromboembolism (VTE) is a condition that consists of two forms of thromboembolic events: deep vein thrombosis (DVT) and pulmonary embolism (PE). It causes a significant loss of life years and increases disability, especially in low- and middle-income countries [1].

The incidence of VTE in the Asian population has been increasing over time, with a rate of 13.8–19.9 per 100,000 persons per year. Despite this increase, the overall burden of VTE in Asia remains lower compared with the Western population because the overall burden of VTE in Asia has been significantly underestimated [2,3]. In Thailand, the number of hospitalizations for VTE has increased from 25,199 cases in 2016 to 32,023 cases in 2020 [4], leading to a higher cost of VTE treatment.

Warfarin is recommended to treat and prevent recurrent VTE [5,6]. However, one major drawback of warfarin is its narrow therapeutic range. Warfarin therapy requires close monitoring of the international normalized ratio (INR) of the prothrombin time [6]. Failure to reach the target INR increases the risk of both thrombotic and bleeding adverse events. Additionally, the food and drug interactions of warfarin are a major concern and can negatively impact patients’ treatment compliance and health-related quality of life [7].

Several direct oral anticoagulants (DOACs) have been introduced to overcome the limitations of warfarin. Clinical evidence shows that DOACs are non-inferior to warfarin in efficacy and have a lower bleeding risk in patients with VTE [8]. Furthermore, treating patients with DOACs does not require close INR monitoring, and there is less concern about food and drug interactions.

The evidence on the effectiveness and safety of DOACs in the Thai population is limited. A real-world retrospective study in Thai patients with atrial fibrillation found that DOACs were associated with a lower rate of all-cause mortality and disease-specific mortality compared with warfarin [9]. Another study examining major bleeding complications between DOACs and warfarin in Thai patients showed that warfarin was linked to intracranial hemorrhage, gastrointestinal bleeding, and an increased risk of death [10]. Even though DOACs come at a much higher price than warfarin, several economic evaluations indicate that DOACs are cost-effective for treating patients with VTE in high-income countries [11,12,13]. However, the cost-effectiveness of DOACs for VTE treatment and prevention in resource-limited settings is scarce and uncertain. Considering a societal perspective, this study aims to evaluate the cost-effectiveness of DOACs compared with warfarin for treating VTE in Thailand. This study will provide information to assist in the policy decision-making process for Thailand’s national essential medication selection process. Additionally, the results of this study can potentially serve as an example for other low- and middle-income countries on the cost-effectiveness of DOACs on VTE.

## 2. Materials and Methods

### 2.1. Overall Description and Study Population

A cost-utility analysis was conducted from a societal perspective to evaluate the effectiveness and cost of anticoagulant treatments for patients diagnosed with VTE. A cohort-based state transition model was designed to mimic VTE disease progression. Eligible patients were diagnosed with VTE and required anticoagulant therapy, with an average age of 64 years [14]. Approximately 53.35% of VTE was diagnosed as PE, while 46.65% was diagnosed as DVT, which was separately modelled [15].

### 2.2. Interventions and Comparator

A total of four DOACs and warfarin as a comparator were assessed. The DOACs available in Thailand were apixaban, rivaroxaban, edoxaban, and dabigatran. The regimens used for VTE treatment for each DOAC were as follows:Apixaban, 20 mg/day for the first 7 days, then 10 mg/day until 6 months. 5 mg/day was selected for patients with recurrence.Rivaroxaban, 30 mg/day for the first 21 days, then 20 mg/day daily for the next 6 months or recurrence.Edoxaban, 60 mg/day in combination with 1 mg/kg enoxaparin every 12 h for the first 5 days, then 60 mg/day for the next 6 months or recurrence.Dabigatran, 300 mg/day in combination with 1 mg/kg enoxaparin every 12 h for the first 5 days, 300 mg/day for the next 6 months or recurrence.

Warfarin was used as a common comparator for VTE treatment. Warfarin was initiated with 1 mg/kg of enoxaparin every 12 h for the first 7 days of treatment, then 3 mg/day was selected. The maintenance dose was based on an average weekly dose with an INR of 2–3 for the Thai population [16]. This study was conducted according to the Consolidated Health Economic Evaluation Reporting Standards 2022 [17].

### 2.3. Model Structure and Assumptions

A Markov simulation model was developed with a lifetime horizon and a 6-month cycle length. The model structure was adapted from previous studies [12,13,18]. It consisted of 9 health states, including VTE on treatment, VTE off treatment, recurrent VTE, gastrointestinal (GI) bleeding, intracranial hemorrhage (ICH), post-ICH, clinically relevant non-major (CRNM) bleeding, chronic thromboembolic pulmonary hypertension (CTEPH), and death. The CTEPH health state was applied only to patients with PE. A schematic is shown in Figure 1.

Patients entered the model based on whether they were diagnosed as DVT or PE at VTE on the treatment health state. Patients who completed 6 months of VTE treatment without adverse events could move to VTE off treatment. Patients with recurrence could move to a recurrence health state, while patients with any adverse events could move to their adverse event health states. The 6-month VTE treatment was based on clinical trials [15,19,20,21,22,23,24].

Several assumptions were made to capture all possible patient trajectories:Patients without adverse events or recurrence were assumed to remain untreated unless they experienced VTE recurrence, and those with recurrence received life-long anticoagulant treatments.Patients with bleeding complications discontinued anticoagulant treatments.CTEPH patients were assumed to be treated with sildenafil, the actual clinical practice for treating CTEPH in Thailand.Patients with PE who experienced bleeding complications were at risk for CTEPH and were assumed to be at the same risk as acute PE patients.Post-thrombolytic syndrome (PTS) was not included due to the limited evidence of PTS in Thailand. In addition, previous studies indicated that PTS had only a small impact on healthcare costs and outcomes [25,26].Patients with non-fatal GI bleeding and ICH were assumed to discontinue anticoagulant treatment permanently. Patients who experienced non-fatal ICH were moved to a post-ICH health state due to an occurrence of disability and remained in this health state until death.

These assumptions were reviewed and agreed upon by a cardiologist and a cardiology pharmacist.

### 2.4. Model Inputs

#### 2.4.1. Efficacy and Transitional Probabilities

A pragmatic search was conducted to determine the treatment effects of each DOAC compared with warfarin. A network meta-analysis was considered first, and landmark randomized controlled trials were considered if no network meta-analysis was available. Briefly, two network meta-analyses were selected for the risks of GI bleeding and ICH [27,28], while the risks of CRNM, VTE recurrence, and VTE-related death were from AMPLIFY [15], RE-COVER II [23], EINSTEIN PE or DVT [21], and Hokusai-VTE studies [24]. The treatment effects and safety of anticoagulant therapy were assumed to be constant over time. The transitional probabilities of VTE patients receiving warfarin were based on previous studies [29,30,31,32,33]. All clinical inputs are presented in Appendix A in the Additional File.

#### 2.4.2. Mortality

The mortality rates were based on previous studies [26,34,35]. The fatal GI bleed rate was derived from the Hokusai-VTE trial, a subgroup studied in East Asian patients [29]. The fatal ICH rate was calculated from national data on stroke outcomes in Thailand [32]. Thai’s age-specific mortality life table was used as the background mortality rate [36]. The excess fatalities from VTE, ICH, and CTEPH were derived from previous studies [34,35,37]. All clinical inputs are presented in Appendix A.

#### 2.4.3. Resource Use and Costs

Drug prices were obtained from the Drug and Medical Supply Information Centre of Thailand [38]. Direct medical costs included costs for diagnostics, outpatient visits, and hospitalization. The model used an average of 3.35 outpatient visits during the 6 months [39]. VTE diagnostic and laboratory monitoring costs were based on current Thai practice. These costs are based on the Thai Standard Costs of Public Health Services [40]. Direct non-medical costs, such as transportation and food costs, were taken from the Thai Standard Cost Lists for Health Technology Assessment [41].

Hospitalization costs for acute PE, DVT, GI bleeding, and ICH were obtained from previous studies using real-world electronic hospital database analyses and adjusted by the inpatient admission rate per year [42,43,44]. The CTEPH cost was estimated using the diagnosis-related group’s relative weight and length of hospital stay [45].

All the costs and resources are listed in Appendix A. Costs were adjusted for inflation to 2021 values using the consumer price index and expressed in Thai Baht [46]. A conversion rate of 31.98 THB/US dollar (USD) was used to convert THB to USD.

#### 2.4.4. Utilities

Utility weights were based on previous studies that utilized the EuroQol-5 Dimension (EQ-5D) or Time Trade-Off (TTO) to estimate patients’ health utilities. A baseline utility was assigned to all patients based on the Thai population’s average score [47]. Patients with CTEPH and post-ICH had lower utilities throughout their lifetime [48,49]. Utility decrements associated with PE, DVT, CRNM bleeds, and anticoagulation were subtracted from baseline utility [48,49,50] (Appendix A).

#### 2.4.5. Model Validation

By simulating the disease progression of VTE patients, the model can provide insights into the impact of different treatment strategies on patient outcomes by tracking changes in each health status over time. A cardiologist and a cardiology pharmacist clinically validated the model and its assumptions during the first expert meeting. A health economist verified the model codes to ensure their validity.

#### 2.4.6. Analyses

A base-case analysis was conducted using the average number of each input, and the incremental cost-effectiveness ratio (ICER) was calculated. Warfarin was used as a common comparison for each DOAC. A fully incremental analysis was also performed to estimate the cost-effectiveness of the next-best alternative. All costs and health outcomes were calculated, and an annual discount rate of 3% was applied to both costs and outcomes. The ICERs of each treatment comparison were compared with a Thai willingness-to-pay (WTP) threshold of THB 160,000/QALY.

A series of one-way sensitivity analyses were performed to explore uncertainties among significant inputs by varying inputs using their 95% confidence intervals (95% CI) or 15% variation. Furthermore, a probabilistic sensitivity analysis (PSA) was conducted using a 10,000-sample Monte Carlo simulation. Each input was assigned a specific probability distribution based on its mean value, and the 95% credible interval was estimated from the PSA. The probabilities of being cost-effective were plotted against the changes in WTP thresholds and displayed in the cost-effectiveness acceptability curves (CEACs).

#### 2.4.7. Budget Impact Analyses

The annual financial impact of adopting DOACs for treating VTE in a Thai healthcare setting was estimated using a budget impact analysis. The budget impact was calculated with a 5-year time horizon and the 6-month medication treatment costs. The incidence of VTE patients and the number of Thai people aged over 60 years were used to estimate the annual number of VTE patients in Thailand. As there were no reports of VTE incidence in Thailand, the incidence of VTE from an Asian country was used [51]. The 5% DOAC uptake rate was assumed to calculate the number of VTE patients receiving DOACs each year. The assumption was consistent with a previous budget impact analysis study [52]. The inputs for the budget impact are presented in Appendix A in the Additional File.

## 3. Results

### 3.1. Number of Clinical Events

Based on the hypothetical cohort of 10,000 VTE patients, all DOACs could help reduce the incidence of VTE recurrence. Edoxaban and apixaban were the most effective DOACs for preventing VTE recurrence compared with warfarin, with 38 fewer cases in edoxaban and 27 fewer cases in apixaban. All DOACs were more effective than warfarin at preventing ICH events. Rivaroxaban had the greatest profile for preventing GI bleeding and ICH, with 106 fewer cases of GI bleeding and 18 fewer cases of ICH than warfarin. Rivaroxaban was the least effective option at preventing CRNM, whereas apixaban resulted in 418 fewer cases of CRNM than warfarin (Table 1).

### 3.2. Base Case Analyses

Compared with warfarin, no DOAC was considered a cost-effective option at the current Thai WTP threshold. Apixaban had the lowest ICER at 269,809 THB ($8437) per QALY gained, followed by dabigatran (707,145 THB or $22,122 per QALY gained), edoxaban (709,944 THB or $22,200 per QALY gained), and rivaroxaban (757,363 THB or $23,682 per QALY gained). Apixaban had the highest QALY of 6.43, or 0.16 incremental QALY, compared with warfarin, while rivaroxaban had the lowest QALY of 6.36, or 0.09 incremental QALY, compared with warfarin. All base-case analysis findings are presented in Table 2. The cost-effectiveness frontier is presented in Figure 2.

Total costs and QALYs obtained for all treatments were applied to generate a cost-effectiveness frontier, which is shown in Appendix A. A fully incremental analysis indicated that apixaban dominated other DOACs with a higher QALY and lower cost. Edoxaban had an ICER of 359,106 THB ($11,229) per QALY gained when compared with rivaroxaban. Dabigatran had an ICER of 597,892 THB ($18,696) per QALY gained compared with edoxaban (Table 3 and Appendix A).

### 3.3. Sensitivity Analyses

One-way sensitivity analysis was performed and displayed as the tornado diagrams in Appendix A. The tornado diagrams depict 10 parameters that had the greatest effects on the ICERs between DOACs and warfarin. The key drivers of ICERs between DOACs and warfarin were the relative risks of VTE-related events and major bleeding complications. The most influential input for apixaban and edoxaban was the risk of GI bleeding, which varied the ICER from 24.46% to less effective and fewer QALYs for apixaban and from 26.89% to 265.48% for edoxaban. The risk of CRNM bleeding was the greatest effect parameter on the ICER in treatment with rivaroxaban and dabigatran, varying between 22.36% and 57.82% for rivaroxaban and between 25.38% and 97.46% for dabigatran. The risk of overall recurrence or VTE-related death in an index PE had the second-largest effect on the ICER of apixaban (14.35–36.84%), rivaroxaban (18.72–48.38%), and dabigatran (17.82–61.34%). The hazard ratio of VTE excess mortality had a moderate effect on the ICER in all DOACs except rivaroxaban. In contrast, the utility decrement during treatment provided a modest impact across all treatments. Costing parameters slightly impacted the ICER, with the percentage changing by less than 2%.

However, no parameter resulted in an ICER of less than 160,000 THB ($5003) per QALY gained. Therefore, all DOACs for VTE treatment were not cost-effective, and the results were robust even when examining all feasible ranges of values for individual parameters.

Probabilistic sensitivity analysis results are shown on cost-effectiveness scatter plots using Monte Carlo simulations (Appendix A) and CEACs. The scatter plot results revealed that the majority of apixaban and rivaroxaban simulations fell in the northeast quadrant, with 95% credible intervals of 183,620 THB ($5742) to 688,222 THB ($21,520) per QALY gained for apixaban and from 415,473 THB ($12,992) to 2,907,062 THB ($90,903) per QALY gained for rivaroxaban (Table 2). The least favorable probabilistic sensitivity distribution occurred with edoxaban and dabigatran. Some simulations fell into the northwest quadrant of the cost-effectiveness plane, which would be more costly and have fewer QALYs than warfarin. At the current WTP, rivaroxaban, edoxaban, and dabigatran have a 0% chance of being cost-effective, and for apixaban, there is only a 0.23% chance of being cost-effective. However, apixaban has the highest possibility of becoming a cost-effective strategy for VTE treatment compared with other DOACs because, at a WTP threshold of 400,000 THB ($12,507), the probability that each DOAC is cost-effective is 85.24% for apixaban and <1% for rivaroxaban, edoxaban, and dabigatran (Figure 3).

### 3.4. Budget Impact Analyses

A total of 4672 VTE patients were expected to receive anticoagulants. Of those, 234 to 1121 patients were expected to be eligible for DOACs for the first to fifth years after DOAC initiation. The budget impact was assessed to increase from 3.27 million THB ($102,251) to 15.71 million THB ($491,244) for apixaban. The total budget impact for the first 5 years of DOAC initiation was 47.47 million THB ($1.48 million) for apixaban. The budget impacts for all DOACs are presented in Table 4.

## 4. Discussion

This study revealed better clinical and economic outcomes with DOACs compared with warfarin in patients with VTE in Thailand. We found that all DOACs could improve QALYs compared with warfarin, ranging from 0.09 to 0.16 QALYs with higher costs of 42,429 to 73,626 THB ($1326 to $2302). All DOACs were not cost-effective options for VTE treatment in Thailand at the current WTP of 160,000 THB/QALY ($5003). Over a patient’s lifetime, apixaban was the most cost-effective option compared with other DOACs. The budget impact was approximately 47.5 million THB ($1.48 million) in the first 5 years of DOAC adoption.

Our findings were inconsistent with previous cost-effectiveness studies, which indicated that DOACs were cost-effective options compared with warfarin for VTE treatment and prevention. Cost-effective studies from Canada and the United Kingdom indicated that apixaban was a cost-effective option for VTE treatment and prevention compared with warfarin [11,12,53]. In contrast, our study showed that DOACs were not cost-effective. The differences might be due to the differences in the healthcare system, treatment costs, and especially the current WTP. To date, our national WTP of 160,000 THB (approximately $5003) has been recommended as the threshold for cost-effectiveness studies in Thailand [54]. It has also been used for several recent cost-effectiveness studies in Thailand [55,56,57]. Compared with the WTP threshold of the United States, which was $50,000–$100,000 per QALY gained, the difference in the WTP threshold might affect the conclusion of this study, which was inconsistent with other studies. However, our findings were in line with the previous studies [11,12,53] that found apixaban to be a dominant option compared with other DOACs.

We also found that all DOACs could reduce the risk of overall VTE recurrence and the risk of bleeding. Patients with apixaban experienced fewer cases of CRNM bleeding than those with other DOACs, while edoxaban was associated with the lowest number of recurrent VTE. These findings were consistent with clinical studies showing that apixaban had the lowest risk of CRNM bleeding compared with other DOACs at a risk ratio of 0.48, while edoxaban had the lowest risk of VTE recurrence in patients with PE [15,21,23,24].

Our one-way sensitivity analyses revealed that the findings of all the DOACs were robust. The major drivers of ICERs were the risk of overall VTE recurrence, VTE-related death, ICH, and GI bleeding. No input could cause the ICER to be below the current Thai WTP, resulting in all DOACs not being cost-effective. The findings were consistent with previous cost-effectiveness studies in terms of the major drivers of the model. Studies from the United States and the Netherlands reported that VTE and major bleeding had the greatest impact on overall treatment costs [58,59,60,61]. Another study from Canada also indicated that CRNM and major bleedings were the major drivers for apixaban to be a cost-effective option among DOACs [53]. A study from the United Kingdom revealed that the relative risk of recurrent VTE could change the ICER of apixaban from being cost-effective [12,18]. Despite the absence of studies on sensitivity analyses in VTE patients in Thailand to compare sensitivity analysis results. A study on the utilization of DOACs for stroke prevention in patients with atrial fibrillation in Thailand indicated that the risk of ischemic stroke and intracranial hemorrhage had a significant impact on the ICERs of all interventions [62]. Another study on patients with non-valvular atrial fibrillation revealed that the hazard ratios of myocardial infarction, intracranial hemorrhage, and ischemic stroke played a crucial role in affecting the ICERs of all DOACs [63]. These studies add further credence to the robustness of the cost-effectiveness findings of DOACs in Thailand through sensitivity analysis and underscore the significance of factors such as the risk of recurrent VTE and VTE-related death as well as the risk of ischemic stroke and intracranial hemorrhage in determining the cost-effectiveness of DOACs.

This study provided valuable information for clinical decision-makers. Although our findings showed that all DOACs were not cost-effective options, they might be needed for VTE patients in some circumstances. According to our findings, apixaban seems to be the most cost-effective option. It provided relatively similar QALYs at a lower cost. When in need, apixaban might need to be first considered. The findings of this study were also valuable for healthcare policymakers. Even though all DOACs are not cost-effective at their current prices, future value-based price negotiations or a managed entry agreement between Thai national policymakers and pharmaceutical companies should be performed to improve the accessibility of DOACs for Thai VTE patients. Additionally, the findings of this study underscore the importance of ongoing monitoring and assessment of the cost-effectiveness of DOACs in Thailand. By conducting regular evaluations, it will be possible to determine any shifts in the efficacy and cost-effectiveness of DOACs, ensuring that the most up-to-date evidence is used to inform treatment decisions for patients with VTE.

Several limitations should be acknowledged. The model did not consider the adjustment of warfarin doses based on INR levels or the adjustment of DOAC doses based on factors such as renal impairment, age, weight, or genetic variations that were not considered in our model. These factors could impact the results, but as we aimed to evaluate the value of DOACs for the general VTE patient population, we believe our findings are still relevant. Secondly, the rates of outcome occurrence were converted to reflect a treatment duration of six months. It was necessary to assume that the outcome rate was consistent throughout the study’s follow-up period. Finally, we assumed that patients with major bleeding discontinued their anticoagulant treatment; however, in real-world practice, they may have switched to other treatment options, which could also affect the results.

## 5. Conclusions

At the current Thai WTP of 160,000 THB/QALY ($5003), none of the DOAC is cost-effective for VTE treatment in Thailand. However, among the DOACs, apixaban was likely to be the most cost-effective treatment. Thai national policymakers might need to consider further price negotiation or establishing a managed entry agreement with pharmaceutical companies to enhance the accessibility of DOACs for patients with VTE in Thailand.

## Figures and Tables

**Figure 1 ijerph-20-03176-f001:**
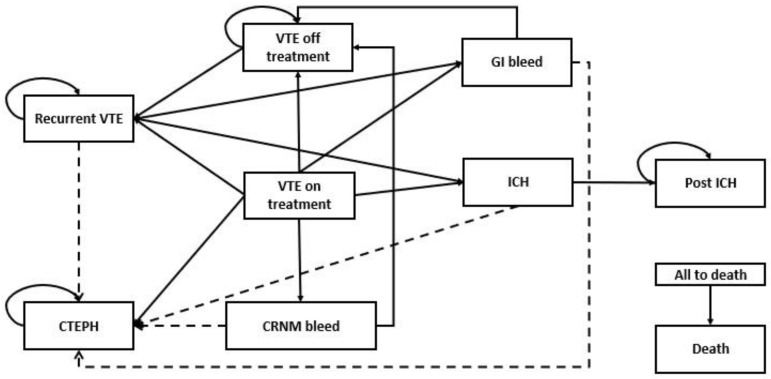
A schematic diagram for a Markov model. Abbreviations: VTE, venous thromboembolism; CTEPH, chronic thromboembolic pulmonary hypertension; CRNM, clinically relevant non-major bleeding; GI, gastrointestinal bleeding; ICH, intracranial hemorrhage.

**Figure 2 ijerph-20-03176-f002:**
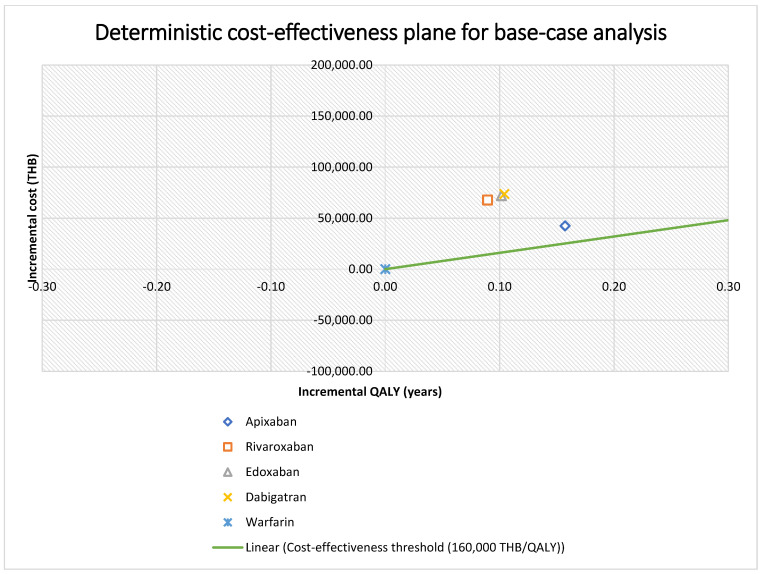
Base-case analysis findings. Abbreviations: THB, Thai bath; QALY, quality-adjusted life year.

**Figure 3 ijerph-20-03176-f003:**
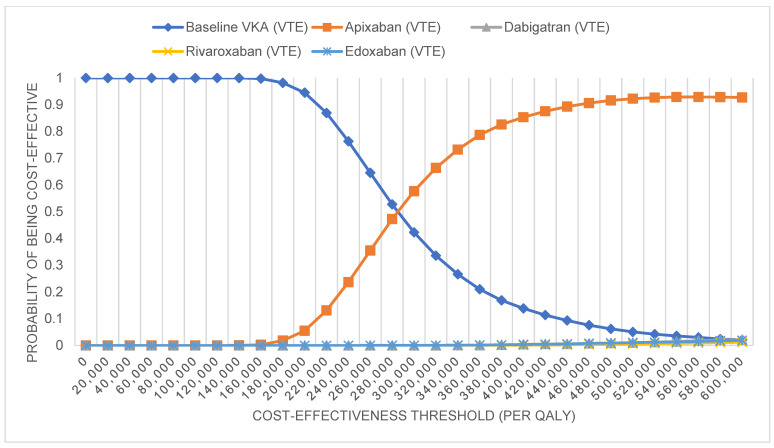
Cost-effectiveness acceptability curve. Abbreviations: VTE, venous thromboembolism; QALY, quality-adjusted life year.

**Table 1 ijerph-20-03176-t001:** Number of clinical events from the simulation of the hypothetical cohort of 10,000 patients diagnosed with VTE.

Number of Clinical Events	Warfarin	Apixaban	Rivaroxaban	Edoxaban	Dabigatran
Recurrent VTE	220	193	219	182	217
GI bleeds	142	142	36	212	200
ICH	20	10	2	6	6
CRNM bleeds	804	386	820	642	498
CTEPH (only PE)	86	86	86	86	86
**Number of clinical events avoided versus warfarin (in a cohort of 10,000 patients)**					
Recurrent VTE	Reference	−27	−1	−38	−3
GI bleeds		0	−106	176	58
ICH		−10	−18	−14	−14
CRNM bleeds		−418	16	−162	−306

Abbreviations: VTE, venous thromboembolism; GI, gastrointestinal; ICH, intracranial haemorrhage; CRNM, clinically relevant non-major bleeds.

**Table 2 ijerph-20-03176-t002:** Base-case cost effectiveness results.

Treatment	Cost (THB)	Life Years	QALYs	Incremental Cost (95% CrI), THB	Incremental QALY (95% CrI)	ICER (95% CrI) THB/QALY Gained	ICER (95% CrI) USD/QALY Gained
Warfarin	129,873	7.41	6.27	Reference			
Apixaban	172,302	7.51	6.43	42,429 (36,522–50,119)	0.16 (0.06–0.22)	269,809 (183,620–688,222)	8437 (5742–21,520)
Rivaroxaban	197,603	7.47	6.36	67,730 (58,438–79,546)	0.09 (0.02–0.16)	757,363 (415,473–2,907,062)	23,682 (12,992–90,903)
Edoxaban	201,944	7.56	6.38	72,071 (62,154–88,027)	0.10 (−0.05–0.17)	709,944 (Dominated–5,252,952)	22,200 (Dominated–164,257)
Dabigatran	203,499	7.47	6.38	73,626 (62,483–86,908)	0.10 (−0.07–0.19)	707,145 (Dominated–5,617,024)	22,122 (Dominated–175,642)

Abbreviations: THB, Thai baht; USD, US dollars (31.98 USD per Thai bath); CrI, credible interval; QALY, quality-adjusted life year; ICER, incremental cost-effectiveness ratio.

**Table 3 ijerph-20-03176-t003:** Fully incremental analysis.

Treatment	Cost (THB)	QALYs	ICER Compared to			
			Lowest Cost (THB) Warfarin	Next Lowest Cost (THB) Apixaban	Relevant Alternative (THB) Rivaroxaban	Relevant Alternative (THB) Edoxaban
Warfarin	129,873	6.274	-	-	-	-
Apixaban	172,298	6.432	269,809	-	-	-
Rivaroxaban	197,638	6.363	757,363	Dominated	-	-
Edoxaban	201,944	6.376	709,944	Dominated	359,106	-
Dabigatran	203,499	6.378	707,145	Dominated	401,389	597,892

Abbreviations: THB, Thai baht; QALY, quality-adjusted life year; ICER, incremental cost-effectiveness ratio.

**Table 4 ijerph-20-03176-t004:** Budget impact results.

Treatment	Year 1	Year 2	Year 3	Year 4	Year 5	Total 5 Years
**Estimated cost of each treatment (THB)**						
**Warfarin**	17,432,498 (USD 528,418)	17,432,498 (USD 528,418)	17,432,498 (USD 528,418)	17,432,498 (USD 528,418)	17,432,498 (USD 528,418)	87,162,491 (USD 528,418)
**Apixaban**	20,406,424 (USD 618,564)	23,816,653 (USD 721,936)	27,423,933 (USD 831,280)	30,696,989 (USD 930,494)	33,970,045 (USD 1,029,707)	134,634,413 (USD 4,081,068)
**Rivaroxaban**	20,877,820 (USD 632,853)	24,150,877 (USD 732,067)	27,423,933 (USD 831,280)	30,696,989 (USD 930,494)	33,970,045 (USD 1,029,707)	137,199,665 (USD 4,158,826)
**Edoxaban**	21,241,025 (USD 643,863)	24,859,125 (USD 753,535)	28,477,226 (USD 863,208)	32,095,327 (USD 972,881)	35,713,427 (USD 1,082,553)	142,386,130 (USD 4,316,040)
**Dabigatran**	21,163,018 (USD 641,498)	24,707,011 (USD 748,924)	28,251,005 (USD 856,351)	31,764,998 (USD 962,868)	35,338,992 (USD 1,071,203)	141,255,023 (USD 4,281,753)
**Budget impact analysis (THB)**						
**Apixaban**	3,273,926 (USD 99,240)	6,384,155 (USD 193,518)	9,494,384 (USD 287,796)	12,604,614 (USD 382,074)	15,714,843 (USD 476,352)	47,471,922 (USD 1,438,979)
**Rivaroxaban**	3,445,322 (USD 104,435)	6,718,379 (USD 203,649)	9,991,435 (USD 302,863)	13,264,491 (USD 402,076)	16,537,547 (USD 501,290)	49,957,174 (USD 1,514,313)
**Edoxaban**	3,808,527 (USD 115,445)	7,426,627 (USD 225,118)	11,044,728 (USD 334,790)	14,662,828 (USD 444,463)	18,280,929 (USD 554,136)	55,223,639 (USD 1,673,951)
**Dabigatran**	3,730,519 (USD 113,080)	7,274,513 (USD 220,507)	10,818,506 (USD 327,933)	14,362,500 (USD 435,359)	17,906,493 (USD 542,786)	54,092,532 (USD 1,639,665)

## Data Availability

The datasets generated and analyzed during the current study are not publicly available due to the confidentiality of data but are available from the corresponding author on reasonable request.

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
