# Peer review of "Economic Evaluation of Direct Oral Anticoagulants Compared to Warfarin for Venous Thromboembolism in Thailand: A Cost-Utility Analysis"

_ijerph, 2023, doi:10.3390/ijerph20043176_

Round 1
Reviewer 1 Report
This is a cost-effectiveness analysis of DOACs compared with warfarin for patients with VTE in Thailand. It is performed using a simulation model, using data from various sources. It found that even though all DOACs showed improved health (QALYs), none of them were considered cost-effective in Thailand at the current WTP-threshold.
I find this study to be of high quality and following best practice. The model is fine, and all data (clinical effects, costs etc) are handled good. It is also good that you provide a budget-impact analysis as a complement.
You obviously fulfil all criteria in CHEERS-2022 checklist, but it might anyway be good to state this.
The results are quite surprisingly as they generally are different from most other studies. Obviously, the main reason is the WTP-threshold being used in Thailand. I think you could discuss these matters more, both from a Thai perspective and from an international perspective. Is the threshold completely strict in Thailand or is flexible depending on severity of disease etc?
Author Response
Reviewer 1
This is a cost-effectiveness analysis of DOACs compared with warfarin for patients with VTE in Thailand. It is performed using a simulation model, using data from various sources. It found that even though all DOACs showed improved health (QALYs), none of them were considered cost-effective in Thailand at the current WTP-threshold.
I find this study to be of high quality and following best practice. The model is fine, and all data (clinical effects, costs etc) are handled good. It is also good that you provide a budget-impact analysis as a complement.
Comment 1
You obviously fulfil all criteria in CHEERS-2022 checklist, but it might anyway be good to state this.
Response 1
Thank You for your support. We added this statement to our method section as shown below.
This study was conducted according to the Consolidated Health Economic Evaluation Reporting Standards 2022
Comment 2
The results are quite surprisingly as they generally are different from most other studies. Obviously, the main reason is the WTP-threshold being used in Thailand. I think you could discuss these matters more, both from a Thai perspective and from an international perspective. Is the threshold completely strict in Thailand or is flexible depending on severity of disease etc?
Response 2
We added some discussion according to WTP threshold as shown below.
Original manuscript
The differences might be due to the differences in the healthcare system, treatment costs, and especially current WTP.
Revised manuscript
The differences might be due to the differences in the healthcare system, treatment costs, and especially current WTP. To date, our national WTP of 160,000 THB (approximately $5,000) has been recommended as the threshold for cost-effectiveness studies in Thailand. It has also been used for several recent cost-effectiveness studies in Thailand. Comparing to the WTP threshold of the United States which was $50,000 – $100,000 per QALY gained, the difference in WTP threshold might affect the conclusion of this study which was inconsistent to other studies.

Reviewer 2 Report
Reviewer report
Title: Economic evaluation of DOACs for venous thromboembolism in Thailand
This paper analysed cost-utility of DOACs compared with warfarin in patients with VTE in Thai context. I have comments as below:
Comments:
1. Title should specify your study design and the comparator used in the analysis.
2. Introduction,
- There is a lack of some important information in this section: How has the situation of VTE in Thailand been? What is its burden in the Thai context? Is there any proof of clinical outcomes of DOACs over warfarin or other treatment for VTE?
- It seems that there was no previous CUA or CEA of DOACs versus warfarin in the context of low- and middle-income countries. Did all of those previous economic evaluations conducted in high-income countries indicate that DOACs were likely cost-effective compared to warfarin? If so, why do Thailand need this study? Or what is the significance of this study?
3. Methods,
- The authors stated that they used a cohort-based state transition model to mimic VTE disease progression in this study. Could the authors please explain more on how you used it and what the additional benefit from this model was?
- Line 142 and 145, what does ‘an additional literature review’ refer to? Please clarify.
- Please check the details of reference #33 which the authors refer to the Thai standard costs of public health services. I cannot find the information of this reference.
- Please cite the reference when you mention to DRG relative weight and LOS, Line 158 Page 4.
- Line 200 Page 5, how did the authors assume the uptake rate of DOAC?
4. Results,
- Small comment from me for this section, in Tables, there are sometime both costing results in THB and USD, some places presented only THB. Please make the currency presentation consistent (either both currencies everywhere or only USD as this is an international journal).
5. Discussion,
- Most of the discussion points referred to (high)cost of DOACs while the one-way sensitivity analysis results indicated a number of clinical outcomes as major drivers of ICERs. Could the authors please discuss more on this with supporting evidence? (especially within the Thai context)
- Please discuss more on policy implication due to the results of this study (apart from price negotiation or MEA in general).
Author Response
Reviewer 2
Comment 1
Title should specify your study design and the comparator used in the analysis.
Response 1
First of all, thank you for taking your valuable time to review and provide feedback on this article.
We revised our title to address the reviewer’s suggestion as shown below.
Original manuscript
Economic evaluation of direct oral anticoagulants for venous thromboembolism in Thailand
Revised manuscript
Economic evaluation of direct oral anticoagulants compared to warfarin for venous thromboembolism in Thailand: A Markov decision analysis
Introduction
Comment 2
There is a lack of some important information in this section: How has the situation of VTE in Thailand been? What is its burden in the Thai context? Is there any proof of clinical outcomes of DOACs over warfarin or other treatment for VTE?
Response 2
There has yet to be a study on the burden of VTE in Thailand. We added important evidence that the overall burden among Asians had been significantly underestimated, and the number of hospitalizations in Thailand has increased yearly based on national admission data. Furthermore, there have been few studies on the efficacy of DOACs in the Thai population. We also included it in the introduction.
We revised the introduction as follows:
Line 55 “The incidence of VTE in the Asian population has been increasing overtime, with an estimate rate of 13.8 – 19.9 per 100,000 person-year. Despite this increase, the overall burden of VTE in Asian remains lower compared to the Western population because the overall burden of VTE in Asia has been significantly underestimated. In Thailand, the number of hospitalizations for VTE has increased from 25,199 cases in 2016 to 32,023 cases in 2020, leading to a higher cost of VTE treatment”
Line 73 “Evidence of the study on the effectiveness and safety of DOACs in the Thai population is limited. A real-world retrospective study in Thai patients with atrial fibrillation found that DOACs were associated with a lower rate of all-cause mortality and disease-specific mortality compared to warfarin. Another study examining major bleeding complications between DOACs and warfarin in Thai patients showed that warfarin was linked to intracranial hemorrhage, gastrointestinal bleeding, and an increase risk of death.”
Comment 3
It seems that there was no previous CUA or CEA of DOACs versus warfarin in the context of low- and middle-income countries. Did all of those previous economic evaluations conducted in high-income countries indicate that DOACs were likely cost-effective compared to warfarin? If so, why do Thailand need this study? Or what is the significance of this study?
Response 3
This study can inform the policymakers for the national essential medication process in Thailand, where CEA is an important piece of evidence required in the medication selection process.
The last paragraph of the introduction has been revised:
Line 81: “Even though DOACs come at a much higher price than warfarin, several economic evaluations indicate that DOACs are cost-effective for treating patients with VTE in high-income countries. However, the cost-effectiveness of DOACs for VTE treatment and prevention in resource-limited settings is scarce and uncertain. This study aims to evaluate the cost-effectiveness of DOACs compared to warfarin for treating VTE in Thailand, considering a societal perspective. This will provide information that will assist in the policy decision-making process for Thailand's national essential medication selection process. Additionally, the results of this study can potentially serve as an example for other low-and-middle-income countries on the cost-effectiveness of DOACs on VTE.”
Methods
Comment 4
The authors stated that they used a cohort-based state transition model to mimic VTE disease progression in this study. Could the authors please explain more on how you used it and what the additional benefit from this model was?
Response 4
The additional benefit of this model has been added to the model validation section.
Line 174 “By simulating the disease progression of VTE patients, the model can provide insights into the impact of different treatment strategies on patient outcome by tracking changes in each health status over time. The model and its assumptions were clinically validated by a cardiologist and a cardiology pharmacist during the first expert meeting. The model codes were verified by a health economist to ensure their validity.”
Comment 5
Line 142 and 145, what does ‘an additional literature review’ refer to? Please clarify.
Response 5
Additional references have been included.
Comment 6
Please check the details of reference #33 which the authors refer to the Thai standard costs of public health services. I cannot find the information of this reference.
Response 6
The website has been added. The information was from the Royal Thai Gazette.
- Ministry of Public Heath, Thailand. (2019). "Thai Standard Costs of Public Health Services”
Comment 7
Please cite the reference when you mention to DRG relative weight and LOS, Line 158 Page 4.
Response 7
The reference has been added.
Comment 8
Line 200 Page 5, how did the authors assume the uptake rate of DOAC?
Response 8
The assumption rate is based on a previous budget impact analysis study in Thailand.
The budget impact analyses section has been rewritten, and the reference has also been added.
Line 199: The 5% uptake rate assumption was also consistent with a previous budget impact analysis study.
Results,
Comment 9
Small comment from me for this section, in Tables, there are sometime both costing results in THB and USD, some places presented only THB. Please make the currency presentation consistent (either both currencies everywhere or only USD as this is an international journal).
Response 9
We added USD into these tables.
Discussion
Comment 10
Most of the discussion points referred to (high)cost of DOACs while the one-way sensitivity analysis results indicated a number of clinical outcomes as major drivers of ICERs. Could the authors please discuss more on this with supporting evidence? (especially within the Thai context)
Response 10
Supporting evidence have been included into this section.
Line 300: Despite the absence of studies on sensitivity analyses in VTE patients in Thailand to draw comparison on sensitivity analysis results. A study on the utilization of DOACs for stroke prevention in patients with atrial fibrillation in Thailand indicated that the risk of ischemic stroke and intracranial hemorrhage had a significant impact on the ICERs of all interventions. Another study on patients with non-valvular atrial fibrillation revealed that the hazard ratios of myocardial infraction, intracranial hemorrhage, and ischemic stroke played a crucial role in affecting the ICERs of all DOACs. These studies add further credence to the robustness of the cost-effectiveness findings of DOACs in Thailand through sensitivity analysis and underscore the significance of factors such as the risk of recurrent VTE and VTE-related death, as well as the risk of ischemic stroke and intracranial hemorrhage in determining the cost-effectiveness of DOACs.
Comment 11
Please discuss more on policy implication due to the results of this study (apart from price negotiation or MEA in general).
Response 11
The policy implication has been revised as follow:
Line 332: “Additionally, the findings of this study underscore the importance of ongoing monitoring and assessment of the cost-effectiveness of DOACs in Thailand, By conducting regular evaluations, it will be possible to determine any shifts in the efficacy and cost-effectiveness of DOACs, ensuring that the most up-to-date evidence is used to inform treatment decisions for patients with VTE”

Reviewer 3 Report
Direct oral anticoagulants have been introduced to overcome warfarin. The study informed that at current willingness to pay in Thailand, Apixban is likely to be the best option for direct oral anticoagulants. Good study by the authors.
Author Response
Reviewer 3
Direct oral anticoagulants have been introduced to overcome warfarin. The study informed that at current willingness to pay in Thailand, Apixban is likely to be the best option for direct oral anticoagulants. Good study by the authors.
Response to reviewer 3
We would like to thank for your support. We really appreciate it.

Round 2
Reviewer 2 Report
I am appreciated with this revision. However, only thing that I think the authors should re-think is about the study design presented in the title of this paper. I prefer the common term such as cost-utility analysis rather than markov decision analysis.
Author Response
Reviewer 2
Comment#1
I am appreciated with this revision. However, only thing that I think the authors should re-think is about the study design presented in the title of this paper. I prefer the common term such as cost-utility analysis rather than Markov decision analysis.
Response#1
We changed the title as shown below
Original manuscript
Economic evaluation of direct oral anticoagulants compared to warfarin for venous thromboembolism in Thailand: A Markov decision analysis
Revised manuscript
Economic evaluation of direct oral anticoagulants compared to warfarin for venous thromboembolism in Thailand: a cost-utility analysis